# Content of Toxic Elements (Arsenic, Cadmium, Mercury, Lead) in Eggs from an Ethically Managed Laying Hen Farm

**DOI:** 10.3390/ani14071133

**Published:** 2024-04-08

**Authors:** Alessandro Guerrini, Paola Roncada, Khaled Mefleh Al-Qudah, Gloria Isani, Fausto Pacicco, Mariantonietta Peloso, Luca Sardi, Doriana Eurosia Angela Tedesco, Gianluca Antonio Romeo, Elisabetta Caprai

**Affiliations:** 1Department of Environmental Science and Policy, University of Milan, Via Celoria 10, 20133 Milan, Italy; alessandro.guerrini@unimi.it (A.G.); doriana.tedesco@unimi.it (D.E.A.T.); 2Department of Veterinary Medical Sciences, University of Bologna, Via Tolara di Sopra 50, 40064 Ozzano dell’Emilia, Italy; gloria.isani@unibo.it (G.I.); luca.sardi@unibo.it (L.S.); 3Department of Veterinary Clinical Sciences, Jordan University of Science and Technology, Irbid 22110, Jordan; alqudah@just.edu.jo; 4Department of Economics, LIUC Cattaneo University, Via Corso G. Matteotti 22, 21053 Castellanza, Italy; fpacicco@liuc.it; 5Chemical Food Department, Istituto Zooprofilattico Sperimentale della Lombardia e dell’Emilia Romagna ‘Bruno Ubertini’, Via P. Fiorini 5, 40127 Bologna, Italy; m.peloso@izsler.it (M.P.); elisabetta.caprai@izsler.it (E.C.); 6Directorate General for Animal Health and Veterinary Medicinal Products (DGSAF), Italian Ministry of Health (MOH), Office 4, Viale Giorgio Ribotta 5, 00144 Roma, Italy; ga.romeo-esterno@sanita.it

**Keywords:** toxic elements, arsenic, cadmium, lead, mercury, eggs, ethically raised hens, backyard hens, rural farm, food safety

## Abstract

**Simple Summary:**

Eggs are a valuable source of nutrients, including proteins, fats, vitamins, and minerals. Moreover, they are also a healthy, nutritious, and easily digestible food. In the last decade, the rearing of chickens at home has been promoted and spread in several Italian communities and throughout the world as a means of promoting the self-consumption of home-produced food. These interventions have been successful because chickens are friendly and colourful pets, and they are inexpensive to buy and feed. However, backyard chickens are potentially exposed to different types of contaminants, such as heavy metals, pesticides, and other chemicals, due to their free access to the natural environment. This study focuses on the presence of four toxic elements in eggs from free-range hens raised in a large park and organic eggs bought at the supermarket.

**Abstract:**

Domestic chicken farming has been promoted and spread in several Italian municipalities and worldwide as an aid to the self-consumption of domestically produced food. This study investigated the levels of four toxic elements (As, Cd, Hg, and Pb) in eggs from an ethical laying hen farm, comparing the element concentrations with those possibly present in supermarket eggs. A total of 201 eggs, 141 from the farm and produced by different hen genotypes, and 60 from the supermarket, were collected. The levels of the toxic elements were evaluated in the yolk, albumen, and eggshells of all eggs. The results show that the supermarket eggs’ yolk and albumen were more contaminated with lead, compared to the rural eggs. Contrarily, the mean content of arsenic was higher in the albumen and eggshells of the rural eggs, compared to the supermarket eggs. The cadmium content was below the LOQ (0.005 mg/kg) in all samples. The mercury content was below or around the LOQ in all rural eggs. Overall, the supermarket egg albumens were significantly more contaminated than the rural ones. No significant differences were found in quality parameters for both types of eggs. The toxic element values that were detected were in line with other studies in the literature. However, despite the concentrations found not representing a risk to the consumers’ health, the results of this study raise a potential food safety issue, and it would be desirable to set specific MRLs for eggs for consumers’ protection.

## 1. Introduction

In the last decade, domestic chicken farming has been promoted and spread in several Italian municipalities and worldwide as an aid to the self-consumption of home-grown food [1,2]. In addition, chickens are kept for economic reasons, as they are cheap to buy and feed [3,4,5]. Today, rural hen breeding is increasingly being used as a source of eggs in both rural and urban areas, with a widespread belief that home-produced eggs are a healthier, safer, and more ethical and sustainable alternative to commercially purchased eggs, with a better respect for animal welfare. Similarly, the laying hen industry is shifting from cages to alternative housing systems (such as aviaries) to address societal concerns regarding animal welfare, such as the best conditions for the birds’ feathers [6].

Eggs are a valuable source of nutrients, including proteins, lipids, vitamins, and minerals. The nutritional value of eggs is largely dependent on the health status, diet, breed, age of the laying hens, and, ultimately, the environment [7,8,9,10]. Laying hen breeds and their genetic diversity have been reported to have a direct effect on egg quality and composition [11]. Rural hens, due to free access to the natural environment, are potentially exposed to different kinds of contaminants, such as heavy metals, pesticides, and other chemicals. The soil could contain residues derived from human or agricultural practices, while the revitalisation and redevelopment of contaminated land for residential purposes creates the potential for high levels of exposure to contaminants. Indeed, it has been suggested that eggs produced on contaminated soil have the potential to become a significant source of contamination for consumers [12]. For this reason, eggs from backyard hens are considered a good indicator of environmental contamination [13]. Several studies have shown that feed contaminated with toxic elements, such as arsenic, cadmium, mercury, and lead can determine the bioaccumulation in chicken tissues and eggs [3,4,14,15,16,17]. As an example, a high content of lead has been detected in chickens’ muscle, liver, kidneys, and eggs, particularly in industrialized regions [18], posing a potential threat to public health. Chickens are exposed to lead, mainly from contaminated soil, through various behaviours such as foraging, roosting, grooming, fighting, and dust bathing, as well as their diet. Lead can also be present in chicken feed and supplements, particularly in worms and invertebrates, bone meal, and oyster shells [19]. The exposure of chickens to lead at a concentration of 1.0 mg/kg in their diet can cause a significant reduction in their growth and in their blood levels of delta-aminolevulinic acid dehydratase, which affects heme production. The chronic intake of lead is associated with high blood concentrations and lead deposition in different tissues and eggs. The highest lead content was found in chickens’ bones followed by the kidneys and liver, while the lowest content was found in the skeletal muscle [17]. Clinical signs of acute lead poisoning in chickens include muscle weakness, ataxia, and loss of appetite, followed by marked weight loss and the cessation of egg production [20].

Human exposure to toxic elements occurs mainly through the oral ingestion of contaminated water and food. Arsenic (As) is a naturally occurring element that is widely distributed in the environment and is stored in the liver, kidneys, heart, and lungs [21,22]. Oxidative stress is one of the mechanisms that is widely accepted to explain arsenic-induced toxicity [23]. Cadmium (Cd) can accumulate in various organs and tissues over time, particularly in the kidneys and liver [21]. Cadmium increases hepatic lipid peroxidation, mitochondrial lipid peroxidation, microsomal lipid peroxidation, and glutathione depletion [24]. Mercury readily binds to the sulfhydryl groups of enzymes and can increase the amount of reactive oxygen species [25]. Lead (Pb) is a neurotoxic metal and is probably carcinogenic to humans, as recognised by the International Agency for Research on Cancer [26]. In children, even low levels of lead exposure can cause developmental delays, a reduced intelligence quotient (IQ), learning difficulties, and behavioural problems [21,27,28,29]. Lead induces oxidative stress by generating reactive oxygen species (ROS) and impairing the body’s antioxidant defence systems [30].

Data on the levels of toxic elements in eggs from rural hens are limited and often lead to conflicting conclusions. This study aimed to: (1) determine the content of four toxic elements (arsenic, cadmium, mercury, and lead) in backyard hen eggs collected from the same rural poultry farm in Puglia (Italy); (2) highlight whether the five different genotypes of rural hens could influence the content of toxic elements in eggs; (3) compare the results with the content of the same elements in organic commercial eggs sold in supermarket.

## 2. Materials and Methods

### 2.1. Sampling

The rural poultry farm was located in Puglia (Italy), extended into a vast park (25,000 square meters), and was designed to provide the hens with the greatest possible comfort while replicating their natural and wild way of life, with the possibility of scratching and free access to the park during the day. The poultry houses were made of wood and utilized by the laying hens for resting at night and laying eggs. The feeding program consisted of a commercial diet based on the nutritional requirements of laying hens, offered ad libitum, and the eggs produced were sold directly to the consumers. According to the farm’s practice, the laying hens were reared until the end of their natural life cycle, not being sent to slaughter at the end of their productive cycle at approximately 52 weeks of age.

In this rural farm, 141 eggs (R group) were collected from 5 different genotypes of laying hens, both pure breeds and hybrids (Araucana, Leghorn, Warren Brown hybrid, Marans, and Olive Egger hybrid), which produced eggshells of different colours: Araucana (blue eggshell, R1; *n* = 30), Leghorn (white eggshell, R2; *n* = 30), Warren Brown hybrid (brick-coloured eggshell, R3; *n* = 21), Marans (chocolate eggshell, R4; *n =* 30), and Olive Egger hybrid (olive-green eggshell, R5; *n* = 30). From each genotype, 30 eggs were collected on the same day. All eggs were processed on the 7th day after deposition.

To compare rural eggs with industrial ones, 12 organic eggs of category “A” were purchased from five different supermarkets (S group), for a total of 60 eggs. All the eggs were immediately sent to the laboratory and analyzed on the 7th day after deposition. European Regulation 2022/2258 extends the expiration date for fresh eggs of category “A” to 28 days after laying.

The experiment was carried out in full accordance with the European legal requirements for the protection of hens on farms [31]. Hens were not subjected to any experimental invasive procedure in vivo. The egg collection was carried out by the farm staff. For these reasons, this research did not fall within the field of application of the Directive 2010/63/EU [32] on the protection of animals used for scientific purposes and, therefore, did not require specific authorization by the local animal welfare and ethical review body.

### 2.2. Egg Quality Parameters

Eggs were weighed and broken on a flat surface. The eggshell thickness was measured using a digital calliper (0.001 mm) (Mitutoyo, Tokyo, Japan) at the equatorial level of each egg. The height of the thick albumen was measured using a tripod micrometre. The Haugh unit, which indicates the freshness of the egg, was calculated using the following formula:HU = 100 × log_10_ (h − 1.7w^0.37^ + 7.6)
where: HU = Haugh unit; h = observed height of the albumen in millimeters; w = weight of egg in grams.

The yolk was separated from the albumen and then weighed. Eggshells were dried at 60 °C for 3 days and then weighed. The weight of the albumen was calculated as the difference between the weight of the egg and the weight of the yolk plus the eggshell. All samples were freeze-dried and stored at −20 °C until analysis. Yolk and albumen were analyzed for their chemical composition parameters, including moisture (950.46), ash (942.05), lipid (960.39), and protein (981.10) content, according to the AOAC 2019 standard [33]

The yolk colour was measured by a Chroma meter CR 400 colourimeter (Minolta Co., Ltd., Osaka, Japan) using the CIE scale [34]. In the Commission Internationale d’Eclairage (CIE) L, a*, b* colour system, the colour is expressed as: L, a*, and b* reflect lightness (0 = black, 100 = white), redness (−100 = green, 100 = red), and yellowness (−100 = blue, 100 = yellow), respectively. All measurements were performed three times and the final value was calculated as the average of the three values ± standard deviation (SD).

### 2.3. Toxic Element Analysis

Eggshell, albumen, and yolk were analyzed separately to evaluate the content of As, Cd, Hg, and Pb. Three samples of up to 5 g of egg albumen and yolk and 0.5 g of eggshell were weighed in polypropylene tubes (Digi-Tubes SCP Science, QuantAnalitica SRL, Osnago, Italy) and subjected to a wet mineralization process with the addition of 10 mL of concentrated nitric acid (J.T. Baker Instra-Analyzed™, Mallinckrodt Baker Inc., Phillipsburg, NJ, USA). Samples were then allowed to cool and heated to 75 ± 10 °C for 750 min overnight. The solution was then cooled, made up to volume with ultrapure deionised water (Evoqua WaterTechnologies, Pittsburgh, PA, USA), and filtered through paper filters, after which 1 mL of the resulting solution was diluted to 10 mL with a dilution solution, an aqueous solution of 2% (*v*/*v*) nitric acid, and 0.5% (*v*/*v*) hydrochloric acid (Suprapur, Sigma-Aldrich, Saint Louis, MO, USA). Analysis was performed by inductively coupled plasma-mass spectrometry (ICP-MS 7700 Series Agilent Technologies Inc. Santa Clara, CA, USA) using an ASX-500 CETAC autosampler (Cetac Technologies, Omaha, NE, USA). The operating parameters were as follows: RF power 1.55 kW, plasma gas flow 15 L/min (Argon-Ar), carrier gas flow 1.01 mL/min (Argon-Ar), and cell gas flow 5 mL/min in ‘He’ mode (low flow) and 10 mL/min in ‘HeHe’ mode (high flow). The isotopes (*m*/*z*) monitored in “He” mode were Hg202 and Pb208, while As75 was monitored in “HeHe” mode. Concentrations were calculated using solvent calibration curves and calibration standards provided by Agilent Technologies Inc. (Santa Clara, CA, USA). The reference materials were prepared using the same acid dilution solution. For each series of analyses, a calibration curve from 0.01 to 100 ng/mL was analyzed, and the correlation coefficient had to be equal to or greater than 0.999 for each element subjected to analysis. A mixture of internal standards (code 5183-4681 Agilent Technologies Inc. Santa Clara, CA, USA) was diluted to a concentration of 1 µg/mL with the dilution solution and continuously infused into the ICP MS through a second entry route to quantify the samples. The accuracy of the method was determined using certified reference material. It consists of applying the analytical procedure to a matrix identical to the one under study, containing known and certified amounts of the analyte. Accuracy is determined by comparison of the experimental mean, with an associated confidence interval, with the certified value. The main limitation to the use of this type of approach is the difficulty in finding matrices under study that are certified for the analyte of interest. For this reason, accuracy was determined by analyzing the certified reference material BCR-185R Bovine Liver (Community Bureau of Reference, Geel, Belgium) in each lot. The concentration values of the reference materials were within the confidence interval indicated by the BCR (Brussels, Belgium).

For each series of analyses, a blank sample was mineralized and treated as described above. At the beginning of each measurement cycle, a tuning operation was carried out with a mixture of several elements to verify the accuracy of the identification of the *m/z* ratio values and the accuracy of the instrument.

The Limit of Detection (LOD) was set at 0.002 mg/kg and the Limit of Quantification (LOQ) at 0.005 mg/kg for all matrices. For statistical analysis, values below the LOD or between the LOD and LOQ were indicated with the LOQ value, which was 0.005 mg/kg (upper bound mode). The average content of the elements are expressed in mg/kg wet weight, and reported as mean ± SD.

### 2.4. Statistical Analysis

The assumption of normality and homogeneity of variance were tested using the Shapiro–Wilk and Levene tests, respectively. Since these requirements were not respected for the element content in all matrices, the data were analyzed with the Kruskal–Wallis test and using the Dunn test for post-hoc comparisons. Data on egg quality were analyzed with One-way ANOVA and Tukey HSD as post-hoc tests. Statistical analysis for the data was carried out for each group and type of egg for each metal and quality parameter, using STATA^®^-Statistical Software Package (StataCorp, College Station, TX, USA), version 16 (StataCorp, 2016). The results were considered statistically significant at *p* < 0.05.

## 3. Results

### 3.1. Toxic Element Content

Data on the contents of arsenic, cadmium, lead, and mercury in the egg yolk, albumen, and eggshells are reported in Table 1, Table 2 and Table 3.

### 3.2. Arsenic (As)

In both types of eggs, different contents of arsenic were found in the matrices that were studied. No significant differences were found between the rural and supermarket egg yolks (*p* > 0.05), nor within the groups (Table 1). However, the albumen of the supermarket eggs had a lower mean of arsenic content (*p* < 0.05) compared with the rural eggs (*p* < 0.01) (Table 2).

Significant differences (*p* < 0.05) were found in the eggshells’ arsenic content (Table 3). The supermarket eggshells were found to be less contaminated than the rural eggshells (*p* < 0.05). In the rural eggshells, the R2 type was found to be more contaminated than the others, albeit not significantly (*p* > 0.05).

### 3.3. Cadmium (Cd)

The cadmium content in all matrices was below the LOQ (0.005 mg/kg) in both the rural and supermarket eggs (Table 3).

### 3.4. Mercury (Hg)

In all yolk samples of both groups, the mercury content was below the LOQ (Table 1). Regarding the egg albumen, the content was always lower than the LOQ in rural eggs, while all albumen samples from supermarket eggs showed values > LOQ values (Table 2), except for group S3, where the content was below the LOQ. Specifically, the albumen of S2 was more contaminated than that of S1 and S5 (*p* < 0.05), which were more contaminated than S4 (*p* < 0.05). Overall, the supermarket egg albumens were more contaminated than the rural ones (*p* < 0.05).

In eggshells from rural hens, only 19 of 141 samples had a detectable mercury content: 12 in the R3 group, 4 in R1, 2 in R4, and 1 in R5. However, the content remained lower than that of the supermarket eggs (*p* < 0.01). In supermarket eggs, the content of mercury in the S2 eggshells was higher than in the S1 eggshells (*p* < 0.01).

### 3.5. Lead (Pb)

Lead was found in both types of eggs and in all matrices. The yolk, albumen, and eggshells of the supermarket eggs had a higher content of lead than the rural eggs (*p* < 0.05) (Table 1, Table 2 and Table 3). In the rural egg yolks, R1 was more contaminated compared to all others; furthermore, R2 and R4 were more contaminated compared to R3 (*p* < 0.01).

In the albumen of the rural eggs, R2 and R4 were less contaminated compared to the R1 and R5 types (*p* < 0.05) (Table 2). Some significant differences were found in the albumen of the supermarket eggs, where the albumen of S1 was more contaminated compared with S2, S3, and S4 (*p* < 0.01). At the same time, the albumen of S5 was more contaminated than in S2 and S3 (*p* < 0.01).

### 3.6. Quality Parameters

The weight of the eggs, eggshell, albumen, and yolk, albumen height, HU index, eggshell thickness, and yolk colour were used to determine the quality of the eggs. The mean and standard deviations of the quality parameters are shown in Table 4.

The contents of proteins, lipids, and ash in the egg albumen and yolk are reported in Table 5. The results of the qualitative parameters and chemical characterization of the eggs showed no significant differences between the rural and supermarket eggs (*p* > 0.05).

## 4. Discussion

Consumers are increasingly interested in healthy food obtained from sustainable and more animal-friendly food systems. This study compared commercial organic eggs from supermarkets with eggs from hens ethically reared in a rural system where the animals live according to their natural behaviour. Living outdoors allows them to graze on grass, eat insects, and enjoy the sun, contributing to a balanced diet and improving their overall health.

Maximum Residual Levels (MRLs) have been set for lead, mercury, and cadmium in various foods such as meat, vegetables, and fish, but no MRLs have been set for these elements in eggs for humans [35,36]. Although home-grown food is not subject to any testing or monitoring protocol for the presence of toxic elements, producers and their customers consume such products in large quantities. The European Food Safety Authority (EFSA) has issued several opinions recommending that the presence of certain elements in different types of food should be monitored to reduce the risk of exposure [37,38,39,40,41].

The monitoring of toxic elements is one of the most important aspects of environmental quality and food safety. Toxic elements occur naturally in the environment and their presence is attributed to natural processes such as volcanism or geochemical activities [42]. However, human activities can increase their levels in different environmental compartments, leading to bioaccumulation in the food chain [41]. Table 6 summarizes the contents of arsenic, cadmium, mercury, and lead in hen eggs reported in the scientific literature.

This study compared the contents of arsenic, cadmium, mercury, and lead in eggs from rural laying hens reared in Puglia (Italy) with those detected in organic eggs from supermarkets. An attempt was also made to evaluate possible differences related to the breed/hybrids and farming system of laying hens. In our study, the mean content of arsenic in the albumen and eggshells of rural hens was higher than that determined in supermarket eggs. Regarding the yolk, it is noteworthy that only two of the 60 supermarket eggs analyzed had values above the LOQ. The higher content in the eggs of the rural hens may be related to the possibility of access to an outdoor enclosure compared to the supermarket ones. The yolk of the rural hen eggs was less contaminated (mean value of 0.011 ± 0.03 mg/kg *w*/*w*) than the albumen and eggshells, with mean values of 0.112 ± 0.12 mg/kg *w*/*w* and 0.202 ± 0.21 mg/kg *w*/*w*, respectively, suggesting that these two matrices could be the main bioaccumulation sites of arsenic. In supermarket eggs, the albumen was more contaminated with arsenic than the eggshells, with mean contents of 0.043 ± 0.065 mg/kg *w*/*w* and 0.010 ± 0.004 mg/kg *w*/*w*, respectively. These data are similar to those reported in the literature. The contents of arsenic reported in the literature are in the range of 0.015–0.029 mg/kg for yolk [13,43,44].

For arsenic, the data of supermarket eggs obtained in this study are in the range of or higher than those reported by Van Overmeire and co-workers [3], where commercial industrial eggs analyzed in Belgian countries showed an arsenic content ranging from 0.009 to 0.024 mg/kg. These values are relatively low compared to other data in the literature [45,46,47,48,49]. As an example, in the study by Uluozlu and colleagues [50], the arsenic content in eggs from popular markets was as high as 0.080 mg/kg.

For cadmium, levels below the LOQ were found in all matrices in both types of eggs. Several authors have found concentrations of cadmium below our LOQ [4,5,13,44,50,51,52,53]. In contrast, higher contents were found in eggs sampled from different markets in Bangladesh [54], and in eggs collected from different local markets in Tehran, Iran [43,55], India [56], and Pakistan [57]. The low content/absence of cadmium in eggs may be due to the protective role of metallothionein acting at the ovarian level. This protein can sequester cadmium and, thus, prevent the accumulation of this toxic metal in eggs. In fact, in laying hens injected with cadmium, it has been reported that cadmium accumulates in the follicle walls of the ovaries due to the induction of metallothionein, while the metal was not present in the follicle yolk [58].

Mercury is present in eggs mainly as methylmercury (Me-Hg) [59], and the content correlates with that found in the internal tissues of laying hens [60,61]. Moreover, eggs are an important route of mercury excretion [62,63]. Our study showed that this metal was not detected in the egg yolk of the rural and supermarket eggs. The absence of mercury in the yolk suggests that it does not bioaccumulate in lipid matrices. Samples of albumen and eggshells from the supermarket eggs and only eggshells from the rural eggs had detectable levels of mercury. These data are consistent with those reported by Gonzàlez-Àlvarez and co-workers [64], who suggest that the greater ability of albumen to bind mercury compared to the egg yolk is due to the presence of specific proteins, particularly ovoalbumin. In general, the highest levels were found in supermarket eggs. The values found in this study are very similar to those found in the study of Waegeneers and co-workers [5], where the mean content ranged from 0.003 to 0.004 mg/kg for home-produced eggs collected in autumn and spring, respectively. Very low contents of mercury in the edible parts of eggs were found in eggs from free-range hens held by private owners and in commercial eggs (0.002 mg/kg and 0.001 mg/kg, respectively) [3,4]. Conversely, very high contents of mercury were found in commercial eggs (0.260 mg/kg) and in eggs from home-raised hens (0.173 mg/kg) sampled in Sydney (Australia) [53].

Eggs can accumulate lead in their eggshell, yolk, and albumen, with the highest levels occurring in the eggshell [17]. The higher content of lead found in the eggshell may be due to the ability of lead to substitute calcium in the eggshell’s formation; eggshell calcification uses Ca^2+^ from intestinal absorption and the mobilization of Ca^2+^ from bone reserves. When chickens are exposed to lead, this divalent cation (Pb^2+^) can enter the deposition pathway in the eggshell [65]. Conversely, in the present study, both rural (0.022 ± 0.071 mg/kg *w*/*w*) and industrial (0.026 ± 0.047 mg/kg *w*/*w*) eggshells had a mean lead content lower than those found in the edible parts; as a consequence, the eggshell does not represent the matrix with the most significant accumulation, as reported in other studies [66], probably because the levels of lead are too low to compete with calcium in the calcification process. The levels found in this research are lower than those reported in previous studies on chronic lead poisoning in hens, where the lead content in eggshells ranged from 2.560 mg/kg to 16.180 mg/kg [67]. In this context, it can be hypothesized that different breeding practices of laying hens may influence the lead accumulation in egg matrices. In the future, more detailed studies should be conducted to verify the role of breed and genetics in the accumulation of lead in laying hen eggs. Concerning the edible part, previous studies have reported that little or no lead could be detected in the albumen of eggs, while the yolk often contained much higher levels of lead [17]. On the contrary, our results showed that the mean content of this metal in the supermarket eggs’ albumen (0.688 ± 0.692 mg/kg *w*/*w*) was higher than that in egg yolk (0.186 ± 0.04 mg/kg *w*/*w*).

According to the recent literature, there are conflicting results regarding the contents of lead in hen eggs of different types. Van Overmeire and colleagues [3,4] reported that the content of lead in backyard eggs was 0.069 mg/kg. Eggs from local markets in India contained 0.489 mg/kg [56], while in Pakistan the lead in eggs reached a value of 0.59 mg/kg [57]. Commercial eggs in France contained 0.01 mg/kg of lead [68]. Waegeneers and co-workers [5] reported that the content of lead in eggs from Belgium was 0.1 mg/kg and Spliethoff et al. (2014) found, in eggs from hens reared in community gardens in New York, a value of 0.07 mg/kg [69]. At the same time, Bautista and co-workers [66] reported that the content of lead in eggs collected in California was 0.056–0.970 mg/kg. Rubio and colleagues [70] documented a higher level of toxic metals in eggs from home-reared hens than in eggs from free-range hens in Spain; however, the lead content was lower than that of the other toxic metals. The estimated lead content in the study by Sobhakumari and colleagues [2] was 0.096–0.26 mg/kg per egg pool (yolk and albumen combined). Trampel and co-workers [17] indicated that no measurable amount of lead was found in egg albumen, while it ranged from 20 to 400 ppb (0.02–0.4 mg/kg) in yolk.

**Table 6 animals-14-01133-t006:** Content of As, Cd, Hg, and Pb (mg/kg) detected in hen eggs (albumen, yolk, and eggshell) in different studies.

	As	Cd	Hg	Pb	Reference
**More common breeds and native laying hens in Iran**					[43]
Yolk	0.029	0.018	0.001	0.126	
Albumen	0.029	0.009	0.001	0.183	
Edible part ^1^	0.028	0.014	0.001	Not available	
**Eggs from three husbandry systems in Greece**					[13]
Yolk conventional	0.014	0.001	-	-	
Yolk organic	0.013	0.002	-	-	
Yolk courtyard	0.015	0.001	-	-	
Albumen conventional	0.005	0.001	-	-	
Albumen organic	0.004	0.001	-	-	
Albumen courtyard	0.006	0.001	-	-	
**Eggs from chickens reared in the same poultry farm in Greece**					[44]
Yolk	0.015	0.001	-	-	
Albumen	0.006	0.001	-	-	
**Eggs from popular markets in Turkey**					[50]
Yolk	0.080	0.003	-	0.100	
Albumen	0.100	0.002	-	0.010	
**Eggs collected from different markets in Bangladesh**					[54]
Edible part ^1^	0.300	0.300	-	0.280	
**Eggs collected in different local markets in Iran**					[55]
Edible part ^1^	0.008	0.130	0.070	0.350	
**Edible part ^1^ of eggs collected in Sydney (Australia)**					[53]
Commercial chicken eggs	0.019	0.001	0.260	0.007	
Home chicken eggs	0.069	0.011	0.173	0.300	
**Eggs collected from four wholesale markets in Bangladesh**					[51]
**Layer** (Edible part ^1^)	0.040	0.010	0.010	0.100	
**Local** (Edible part ^1^)	0.040	0.010	0.010	0.240	
**Organic** (Edible part ^1^)	0.040	0.010	0.010	0.060	
**Free-range hens’ eggs collected in home-producing farms near illegal dumping of industrial wastes in Italy**					[52]
Edible part ^1^	-	-	0.011	<LOQ	
**Home-produced eggs in Belgium**					[5]
Edible part ^1^					
**autumn**	<LOQ	0.001	0.003	0.116	
**spring**	<LOQ	<LOQ	0.004	0.074	
**Eggs from free-range hens held by private owners (PO) and from commercial egg production farms in Belgium**					[3,4]
Edible part ^1^					
Private owners	0.016	0.001	0.002	0.069	
Commercial farms	0.013	0.001	0.001	0.009	
**Fresh eggs from caged hens in Tenerife (Spain)**					[71]
Yolk	-	<LOQ	-	0.040	
Albumen	-	<LOQ	-	0.020	
Edible part ^1^	-	<LOQ	-	0.020	
**Eggs from free-range hens in Tenerife (Spain)**					[70]
Yolk	-	<LOQ	-	0.060	
Albumen	-	<LOQ	-	0.040	
Edible part ^1^	-	<LOQ	-	0.030	
**Eggs from home-grown hens in Tenerife (Spain)**					
Yolk	-	<LOQ	-	0.090	
Albumen	-	<LOQ	-	0.050	
Edible part ^1^	-	0.002	-	0.050	
**Eggs collected in New York City (USA)**					[69]
Henhouses	-		-	0.017	
Rural	-		-	<LOQ	
Store-bought	-		-	<LOQ	
**Eggs collected in Ames, Iowa (USA)**					[17]
**Eggs from hens observed eating lead paint**					
**Eggshell**					
after 1 day	-	-	-	0.078	
after 5 days	-	-	-	0.090	
after 9 days	-	-	-	0.185	
control	-	-	-	0.017	
**Yolk**					
after 1 day	-	-	-	0.170	
after 5 days	-	-	-	0.187	
after 9 days	-	-	-	0.262	
**Control**	-	-	-	0.012	
**Eggs collected in Israel**					[67]
Eggshell (min–max)				0.43–1.40	
Yolk (min–max)				0.08–0.18	
**Two different hen husbandry systems in Romanian regions and Greece**					[72]
Backyard					
Yolk (min–max)	-	-	-	0.030–0.170	
Albumen (min–max)	-	-	-	0.001–0.590	
Barn					
Yolk (min–max)	-	-	-	0.020–0.060	
Albumen (min–max)	-	-	-	0.002–0.060	
**Eggs from Lower Hunter, NSW, Australia**					[12]
Home-grown eggs	<LOQ	<LOQ	-	0.090	
Commercial eggs	<LOQ	<LOQ	-	0.040	
**Hens’ eggs in India**	-	0.072	-	0.489	[56]
**Poultry eggs samples collected in Pakistan**					[57]
Farm	-	0.074	-	0.584	
Market	--	0.072	-	0.602	
**Eggs samples collected in California (USA)**					[66]
Edible part ^1^ (min–max)	-	-	-	0.056–0.970	
Eggshell (min–max)	-	-	-	0.075–1.8	
**As content in eggs from Taiwan**	0.025–0.026	-	-	-	[46]
**As content in eggs from Jiangsu Province (China)**	0.007	-	-	-	[49]
**Brazilian egg samples**					[47]
Conventional eggs	0.023	0.003	-	0.011	
Home-produced eggs	0.022	0.001	-	0.089	
**Eggs collected in Campania (Italy)**					[73]
Edible part ^1^	0.007	0.003	-	0.019	
**Eggs collected in Bangladesh**					[45]
Edible part ^1^	0.30	-	-	-	
**Eggs randomly obtained in markets of Spain**					[48]
Edible part ^1^	0.015	0.008	0.001	0.015	
**Studies of Lead and Mercury in Laying Hens (Korea)**					[15]
Week 4-Control	-	-	ND	ND	
Week 8-Control	-	-	ND	ND	
**Backyard chicken layer flocks in California (different locations)**					[2]
Egg pool				0.096–0.260	
**Italian rural and industrial eggs**					This research
**Rural eggs**					
Yolk	0.011	<LOQ	<LOQ	0.089	
Albumen	0.112	<LOQ	<LOQ	0.039	
Eggshell	0.202	<LOQ	0.006	0.022	
**Supermarket eggs**					
Yolk	0.006	<LOQ	<LOQ	0.187	
Albumen	0.043	0.005	0.266	0.688	
Eggshell	0.010	0.009	0.009	0.052	

Averages of the concentrations are shown. Only if averages are not available, min–max data are reported. A summary of the peculiarities of the published work is provided in the first column. ^1^ “Edible part” means yolk plus albumen mixed and analyzed. - means that the chemical element was not the subject of the research. ND: not detected.

Finally, the results of the qualitative parameters and chemical characterization showed no significant differences between the rural and supermarket eggs; this evidence is in agreement with other studies [74,75,76,77]. Therefore, the current results suggest that the presence of metals in egg components does not influence the quality and chemical parameters of eggs.

## 5. Conclusions

Interest in and sensibility to animal health and welfare and the consumption of more sustainable food has led to an increase in the popularity of backyard hen farming as a source of eggs in rural and urban areas. This approach to poultry farming is based on the idea that home-raised hen eggs are a healthier, safer, and more sustainable alternative to commercial eggs. There is no doubt that hen eggs are an important component of the human diet worldwide, consumed as they are or in different food preparations (pasta, cakes, biscuits, etc). Various studies have assessed the presence of environmental contaminants in eggs, the exposure to which may pose certain health risks to consumers. However, backyard hen farmers are not legally bound to rules concerning the quality of their eggs and/or the way that their farming system is managed (types of house, type of external paddock, diet, etc.), except for the basic rules of public health (surveillance of avian influenza etc.). In these systems, compared to the industrial reality where the hens are reared in a protected environment, the soil and the poultry house can be a source of heavy metals which eventually accumulate in the eggs through soil–egg transfer.

The results of this research must be considered with some limitations. Firstly, the study was carried out on a single laying hen farm due to the specific and rare characteristics of laying hens for egg production. Secondly, the eggs analysed came from hens of different ages because they were ethically reared, respecting their normal life cycle, and are not slaughtered at the end of the production cycle. Finally, the analysis of toxic elements in feed can be difficult in this type of farming because hens eat what they find in the environment and this is subject to many factors, mainly related to seasonality. Similarly, it is difficult to assess the levels of toxic elements in the feed of hens producing organic eggs for sale in supermarkets. However, future studies could consider sampling soil and grass in the park where the hens feed, taking into account seasonal changes in feed.

In conclusion, even if the metal content detected in this study in both types of eggs is generally low and probably does not pose a risk to the consumers, these heavy metals may accumulate in the human body, raising potential public health problems in the case of the frequent consumption of eggs containing toxic elements; this should be taken into consideration when setting MRLs for eggs. Moreover, consumers of free-range or organic eggs should be aware of the potential health risks associated with the consumption of these types of eggs, considering also the overall intake of potential contaminants in their diet with the consumption of other foods.

## Figures and Tables

**Table 1 animals-14-01133-t001:** Contents of As, Cd, Hg, and Pb in the egg yolk of rural (R) eggs from different laying hen genotypes and supermarket eggs (S). The data are expressed as mg/kg wet weight and reported as mean ± SD, N is the sample size.

		Metals (mg/kg)			
Egg Type	N	As	Cd	Hg	Pb
R1 *	30	0.009 ± 0.00	<LOQ	<LOQ	0.141 ± 0.53 ^a^
R2	30	0.010 ± 0.00	<LOQ	<LOQ	0.117 ± 0.03 ^b^
R3	21	0.006 ± 0.00	<LOQ	<LOQ	0.023 ± 0.01 ^c^
R4	30	0.007 ± 0.00	<LOQ	<LOQ	0.083 ± 0.05 ^b^
R5	30	0.024 ± 0.07	<LOQ	<LOQ	0.061 ± 0.09 ^bc^
RM	141	0.011 ± 0.03	<LOQ	<LOQ	0.089 ± 0.25 ^y^
S1 ^ǂ^	12	<LOQ	<LOQ	<LOQ	0.203 ± 0.03
S2	12	0.012 ± 0.02	<LOQ	<LOQ	0.181 ± 0.05
S3	12	<LOQ	<LOQ	<LOQ	0.186 ± 0.06
S4	12	<LOQ	<LOQ	<LOQ	0.181 ± 0.03
S5	12	<LOQ	<LOQ	<LOQ	0.181 ± 0.03
SM	60	0.006 ± 0.01	<LOQ	<LOQ	0.186 ± 0.04 ^x^

* R1, Araucana breed (blue eggshell); R2, Leghorn breed (white eggshell); R3, Warren Brown hybrid (brick-colored eggshell); R4, Marans breed (chocolate eggshell); R5, Olive Egger hybrid (olive-green eggshell); RM, mean of rural eggs. ^ǂ^ S1, S2, S3, S4, and S5, eggs from the five supermarkets; SM, mean of supermarket eggs. ^a,b,c^, means with different superscripts (^a,b,c^) in the same column are significantly different (*p* < 0.05). ^x,y^ overall means with different superscripts (^x,y^) in the same column are significantly different (*p* < 0.05).

**Table 2 animals-14-01133-t002:** Contents of As, Cd, Hg, and Pb in the egg albumen of rural eggs (R) from different laying hen genotypes and supermarket eggs (S). The data are expressed as mg/kg wet weight and reported as mean ± SD, N is the sample size.

				Metals (mg/kg)	
Egg Type	N	As	Cd	Hg	Pb
R1 *	30	0.137 ± 0.16	<LOQ	<LOQ	0.069 ± 0.14 ^a^
R2	30	0.094 ± 0.10	<LOQ	<LOQ	0.010 ± 0.01 ^b^
R3	21	0.097 ± 0.12	<LOQ	<LOQ	0.027 ± 0.04 ^ab^
R4	30	0.116 ± 0.11	<LOQ	<LOQ	0.014 ± 0.02 ^b^
R5	30	0.113 ± 0.11	<LOQ	<LOQ	0.070 ± 0.18 ^a^
RM	141	0.112 ± 0.12 ^x^	<LOQ	<LOQ ^y^	0.039 ± 0.11 ^y^
S1 ^ǂ^	12	0.033 ± 0.02	<LOQ	0.095 ± 0.03 ^b^	1.495 ± 0.44 ^a^
S2	12	0.039 ± 0.02	<LOQ	1.077 ± 3.70 ^a^	0.090 ± 0.13 ^c^
S3	12	0.042 ± 0.03	<LOQ	<LOQ ^c^	0.034 ± 0.04 ^c^
S4	12	0.057 ± 0.13	<LOQ	0.050 ± 0.04 ^c^	0.550 ± 0.57 ^bc^
S5	12	0.043 ± 0.01	<LOQ	0.101 ± 0.02 ^b^	1.270 ± 0.25 ^ab^
SM	60	0.043 ± 0.06 ^y^	<LOQ	0.266 ± 1.65 ^x^	0.688 ± 0.69 ^x^

* R1, Araucana breed (blue eggshell); R2, Leghorn breed (white eggshell); R3, Warren Brown hybrid (brick-colored eggshell); R4, Marans breed (chocolate eggshell); R5, Olive Egger hybrid (olive-green eggshell); RM, mean of rural eggs. ^ǂ^ S1, S2, S3, S4, and S5, eggs from the five supermarkets; SM, mean of supermarket eggs.^a,b,c^, means with different superscripts (^a,b,c^) in the same column are significantly different (*p* < 0.05). ^x,y^ overall means with different superscripts (^x,y^) in the same column are significantly different (*p* < 0.05).

**Table 3 animals-14-01133-t003:** Contents of As, Cd, Hg, and Pb in the eggshells of rural eggs (R) from different laying hen genotypes and supermarket eggs (S). The data are expressed as mg/kg wet weight and reported as mean ± SD, N is the sample size.

		Metals (mg/kg)			
Egg Type	N	As	Cd	Hg	Pb
R1 *	30	0.179 ± 0.17	<LOQ	0.007 ± 0.00	0.045 ± 0.14
R2	30	0.250 ± 0.25	<LOQ	<LOQ	0.014 ± 0.01
R3	21	0.113 ± 0.16	<LOQ	0.009 ± 0.00	0.012 ± 0.01
R4	30	0.205 ± 0.20	<LOQ	<LOQ	0.014 ± 0.02
R5	30	0.236 ± 0.23	<LOQ	<LOQ	0.021 ± 0.03
RM	141	0.202 ± 0.21 ^x^	<LOQ	0.006 ± 0.00 ^y^	0.022 ± 0.07 ^y^
S1 ^ǂ^	12	0.009 ± 0.00	<LOQ	0.006 ± 0.00 ^b^	0.054 ± 0.09
S2	12	0.009 ± 0.00	<LOQ	0.012 ± 0.00 ^a^	0.025 ± 0.02
S3	12	0.011 ± 0.00	<LOQ	0.008 ± 0.00 ^ab^	0.012 ± 0.00
S4	12	0.009 ± 0.00	<LOQ	0.008 ± 0.00 ^ab^	0.016 ± 0.00
S5	12	0.012 ± 0.00	<LOQ	0.009 ± 0.00 ^ab^	0.155 ± 0.45
SM	60	0.010 ± 0.00 ^y^	<LOQ	0.009 ± 0.00 ^x^	0.052 ± 0.20 ^x^

* R1, Araucana breed (blue eggshell); R2, Leghorn breed (white eggshell); R3, Warren Brown hybrid (brick-colored eggshell); R4, Marans breed (chocolate eggshell); R5, Olive Egger hybrid (olive-green eggshell); RM, mean of rural eggs. ^ǂ^ S1, S2, S3, S4, and S5, eggs from the five supermarkets; SM, mean of supermarket eggs. ^a,b^, means with different superscripts (^a,b^) in the same column are significantly different (*p* < 0.05). ^x,y^ overall means with different superscripts (^x,y^) in the same column are significantly different (*p* < 0.05).

**Table 4 animals-14-01133-t004:** Qualitative parameters (mean ± SD) of rural (R) and supermarket (S) eggs.

Parameter	Type	Mean ± SD
Egg weight (g)	R S	59.2 ± 3.133 62.3 ± 1.298
Egghell weight (g)	R S	7.12 ± 0.377 8.2 ± 0.224
Albumen weight (g)	R S	35.0 ± 2.692 36.3 ± 0.808
Yolk weight (g)	R S	15.8 ± 0.760 16.4 ± 0.505
Albumen height (µm)	R S	6.60 ± 0.453 5.94 ± 0.559
HU Index	R S	80.4 ± 3.415 74.5 ± 4.305
Eggshell thickness (mm)	R S	0.349 ± 0.018 0.395 ± 0.007
Yolk colour		
L	R S	54.1 ± 1.521 54.1 ± 1.695
a*	R S	0.440 ± 1.880 0.620 ± 2.137
b*	R S	41.8 ± 2.235 44.8 ± 3.662

**Table 5 animals-14-01133-t005:** Contents of proteins, lipids, and ash in egg albumen and yolk (mean ± SD) in rural (R) and supermarket (S) eggs. Values are expressed as a percentage.

Albumen Proteins (%)	R S	10.06 ± 0.802 10.61 ± 0.787
Yolk Proteins (%)	R S	16.22± 0.466 16.47 ± 0.628
Yolk Lipids (%)	R S	32.26 ± 0.560 32.81 ± 1.488
Egg white ash (%)	R S	0.70 ± 0.024 0.67 ± 0.050
Yolk ash (%)	R S	1.66 ± 0.089 1.71 ± 0.118

## Data Availability

Data are contained within the article.

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
