# Peer review of "Content of Toxic Elements (Arsenic, Cadmium, Mercury, Lead) in Eggs from an Ethically Managed Laying Hen Farm"

_animals, 2024, doi:10.3390/ani14071133_

Round 1

Reviewer 1 Report

Comments and Suggestions for Authors

Dear authors,

You’ve made extensive work on analyzing lots of egg samples and literature review. Introduction part and methodology are well described and explained, however there are some issues in the Results and Discussion parts. Please find my notes attached.

Best regards

Author Response

Response to Reviewer 1 Comments

The authors would like to thank the reviewer for the time spent in reviewing the manuscript. Your feedback is appreciated and we have attempted to address each comment below.

Line 33-34. „while its (lead) content in the eggshell was higher in rural eggs “– data in Table 3 says different (rural vs. supermarket 0.022 vs. 0.052) – check this

Checked and modified

Line 37. “Mercury in all egg samples was below the level of quantification” – what about Hg in SM eggs (albumen, shell)? – check

Checked and modified

Line 196. Table 1 – must say I've never seen this type of significance labelling. Please use letters a,b,c,d….. This way of labelling is confusing.

- What about R5 value for As (0.024), isn't that value significant and different from other values in this column?

- R1 value for Pb (0.141)???? significance?

We changed the superscripts to identify the significance with the common letters “a, b, c, etc”, and the letters x,y for comparisons between eggs types (rural vs supermarket). Overall, the significances detected were revised to apply other parametric tests such as ANOVA, also suggested by another reviewer. 

Line 206. Table 2 – comment concerning statistics same as line 196.

- What about S2 value for Hg (1.077)????

- What about R5 value for Pb (0.070)???? – isn’t it different from R4 also?

Line 216. Table 3 - comment concerning statistics same as line 196.

- SM value for Hg is marked with SM , it should be RM

Line 253. S4 albumen was more contaminated (P < 0.05) than the S5 albumen – LESS?  

Line 265-268. What about R1? There is no need to repeat values from tables in text

Regarding the comment from Line 206 to line 268, the significances detected were revised to apply other parametric tests such as ANOVA, also suggested by another reviewer. The tables have been modified. 

Line 287. Table 5. Missing content – same data as in Table 4

We are sorry for the mistake. Checked and modified.

Line 323-325. Move this sentence upper in the text, for example in Line 313, before “Mercury (Hg) occurs in both organic”

We have modified the paragraph between lines 303-325 as suggested by Reviewer 3

Line 348-349. For arsenic, the data of supermarket eggs obtained in this study are in the range or lower than those reported by Van Overmeire… As in SM albumen in your research is 0.043

Done

Line 359. markets in Tehran, Iran [41,53]. as well as India [63] and Pakistan [64] – optionally add these references

Done

Line 369-371. Only a few samples of albumens from supermarket eggs and eggshells from rural eggs have a detectable content 370 of mercury. – I wouldn’t say only a few, your average for SM albumen is 0.266 (table 2). – check this

Checked and modified

Line 373-379. Check references throughout this part. Line 373 is reference 4, not 3, according to values and content. Line 376-377 – reference 5-is it, or it’s 4 again?

Checked and modified

Line 392. ranged from 2.560 mg/kg [62] to 16.180 mg/kg [17]. – all of this is from reference 62, remove 17

Checked and modified

Line 405-407. “At the same time, Bautista and co-workers [61], reported that the content of lead in eggs collected from community gardens in New York was similar, between 0.01-0.07 mg/kg.”

- reference 61 is about California and Pb levels 0.056-0.970

- values 0.01-0.07 are from reference 68

Checked and modified

Line 411. …Sobhakumari and colleagues [2] was 3-13 μg/kg (0.003-0.013 mg/kg) per egg pool – these authors say that: “Lead concentrations varied 0.096–0.26 μg/g of egg wet weight. Based on the detected egg lead concentrations and average egg size of 50 g, the estimated lead content was 3–13 μg per egg pool.” - 3-13 μg per egg pool not per kg of egg. If you want to use theirs values, it’s better to use those expressed in μg/g (0.096-0.26) ie. mg/kg

Checked and modified

Line 416. Table 6. Carefully check all references in brackets to ensure that the data in Table are from appropriate reference.

For example, values for Home-produced eggs in Belgium

Edible part1

autumn

<LOQ

0.001

0.003

0.116

[5]

spring

<LOQ

<LOQ

0.004

We thank this reviewer for reviewing the table. We have re-checked the table and made some changes based on the reviewer's suggestions.

Reviewer 2 Report

Comments and Suggestions for Authors

This research studied an interesting topic on heavy metal in eggs. However, here I identify some potential issues:

1. It only compared a limited number of eggs from one rural farm with some commercial eggs. The heavy metal profile could merely be a reflection of that specific rural farm rather than being a representative of overall rural farms.

2. For the egg quality test, the sample size is also too small.

3. The abstract did not fully reflect the four objectives stated in the last paragraph of the introduction.

4. Using a t-test for multiple comparisons can increase the type-I errors. ANOVA could be a better way to run the statistical analysis. If so, the results and discussion might be changed. 

Author Response

Response to Reviewer 2 Comments

The authors would like to thank the reviewer for the time spent in reviewing the manuscript. Your feedback is appreciated and we have attempted to address each comment below.

This research studied an interesting topic on heavy metal in eggs. However, here I identify some potential issues:

  1. It only compared a limited number of eggs from one rural farm with some commercial eggs. The heavy metal profile could merely be a reflection of that specific rural farm rather than being a representative of overall rural farms.

Our aim was to specifically verify the presence of metals in this particular farm, which supplies eggs for direct sale to the consumer and on Italian territory. Furthermore, the purpose of considering this specific farm is that the hens live free range without any particular human intervention and are not intended for direct human consumption at the end of their productive career. Therefore, this survey was not intended to represent the totality of free-range or rural farms.

  1. For the egg quality test, the sample size is also too small.

The purpose of our work was to assess the presence of toxic elements in eggs. We thought it would be useful to add an assessment of egg quality to check that there were no differences between the two types of eggs or that the differences, if any, were not dependent on the presence of toxic elements.

  1. The abstract did not fully reflect the four objectives stated in the last paragraph of the introduction.

The revised abstract includes the main objectives of the study and the most relevant results. The limited number of words available does not allow for further additions.

  1. Using a t-test for multiple comparisons can increase the type-I errors. ANOVA could be a better way to run the statistical analysis. If so, the results and discussion might be changed.

As suggested, the statistical analysis was revised with the application of the ANOVA test and non-parametric test (Kruskal-Wallis). Consequently, we changed the superscripts to identify the significance with the common letters “a, b, c, etc”, and the letters x,y for comparisons between eggs types (rural vs supermarket). Overall, the significances detected were revised, and consequently the results and discussion section. The statistical analysis paragraph was rewritten.

Reviewer 3 Report

Comments and Suggestions for Authors

The manuscript has significant errors that must be corrected before proceeding further in the publication process. The main concerns relate to the statistics used, incomplete key descriptions in the methodology regarding eggs from backyard hens, and the absence of a limitations section. Below are detailed comments:

Major issues:

  1. Statistics: Given the multiple experimental groups, the t-test, which compares only two groups, seems inappropriate for this analysis. Using multiple t-tests increases the Type I error. Additionally, some data appear to lack normal distribution (as indicated by zero SD in tables 1 or 3), making the application of parametric tests also inappropriate. The statistical approach must be corrected.
  2. This study has numerous significant methodological errors, including the absence analysis of toxic elements in the feed given to backyard hens, which undermines reliable comparisons between eggs from different rural hen genotypes, and the lack of soil analyses in the park, which affects the reliability of comparisons with supermarket eggs.
  3. Were all backyard hens the same age? Egg size and other traits, such as eggshell thickness, depend on the hen's age. What was the age of hens when eggs were collected for analysis? Were all backyard hens fed the same diet?
  4. Title: Consider revising the use of "ethically," as this aspect is not discussed in the abstract or introduction, where the study's aim and hypothesis are stated.
  5. L91-92: Consider removing the 4th point, as these aspects are not mentioned in the introduction or title. While these indices are crucial for determining egg quality, they do not seem central to the study's main purpose.
  6. L111: Does the symbol "S" refer to "supermarket" or indicate the size of the purchased eggs? If the former, what was the size of the commercial eggs purchased, and how does it compare to the size of eggs laid by backyard hens?

Other comments:

L49-50: Consider removing this sentence. A non-technical simple summary would be more appropriate.

L103: Does "±12 months" intend to mean "at the age of approximately 52 weeks"?

Be consistent and use the term "eggshell" throughout the study.

L122: Correct the Haugh unit formula to "w^0.37".

L134: Correct the reference format.

For all equipment used, provide the apparatus model number. For equipment and chemicals, include the manufacturer's location. This is mandatory for this journal as it aids in replicating measurements.

L153: This line is unclear; please revise.

L154: There is no isotope as "hg2O2". Carefully revise the isotope notation – Pb208 vs 75As.

L162: Change "part number" to "code"; standards are not part of the apparatus. Also, this standard appears to have a concentration of 10 μg/mL, not 1 μg/mL; please verify.

The title of Table 6 is misleading, as no comparison with other studies is provided.

L303-325: Try to shorten this paragraph. Suggestion: Summarize the toxicological effects and exposure limits of all analyzed toxic elements in a table.

Author Response

Response to Reviewer 3 Comments

The authors would like to thank the reviewer for the time spent in reviewing the manuscript. Your feedback is appreciated and we have attempted to address each comment below.

The manuscript has significant errors that must be corrected before proceeding further in the publication process. The main concerns relate to the statistics used, incomplete key descriptions in the methodology regarding eggs from backyard hens, and the absence of a limitations section. Below are detailed comments:

Major issues:

  1. Statistics: Given the multiple experimental groups, the t-test, which compares only two groups, seems inappropriate for this analysis. Using multiple t-tests increases the Type I error. Additionally, some data appear to lack normal distribution (as indicated by zero SD in tables 1 or 3), making the application of parametric tests also inappropriate. The statistical approach must be corrected.

As suggested, the statistical analysis was revised with the application of the ANOVA test (and Tukey HSD post-hoc test), and non-parametric test (Kruskal-Wallis). Previously, the assumption of normality and homogeneity of variance were tested using the Shapiro-Wilk and Levene tests, respectively.

Consequently, we changed the superscripts to identify the significance with the common letters “a, b, c, etc”, and the letters x,y for comparisons between eggs types (rural vs supermarket). Overall, the significances detected were revised, and consequently the results and discussion section. The statistical analysis paragraph was rewritten.

  1. This study has numerous significant methodological errors, including the absence analysis of toxic elements in the feed given to backyard hens, which undermines reliable comparisons between eggs from different rural hen genotypes, and the lack of soil analyses in the park, which affects the reliability of comparisons with supermarket eggs.

The analysis of toxic elements in hen feed is very difficult to carry out as hens eat what they find in their environment and this is subject to many factors, mainly related to seasonality. In future work, sampling of soil and grass in the park could be considered. However, as the park is very large and the hens are free to move throughout the available area, the number of samples must be representative of the whole environment, taking into account seasonal changes.

  1. Were all backyard hens the same age? Egg size and other traits, such as eggshell thickness, depend on the hen's age. What was the age of hens when eggs were collected for analysis? Were all backyard hens fed the same diet?

The age of the backyard hens varies, because they are ethically reared, respecting their normal life cycle, and are not slaughtered. Hens eat what they find in their environment, so there are no differences in diet.

  1. Title: Consider revising the use of "ethically," as this aspect is not discussed in the abstract or introduction, where the study's aim and hypothesis are stated.

In the first lines of the introduction we have included the term etically because it implies that the way rural hens are reared is more ethical and welfare-friendly. We have included a more comprehensive description in the materials and methods. We have included a fuller description in the materials and methods section.

  1. L91-92: Consider removing the 4th point, as these aspects are not mentioned in the introduction or title. While these indices are crucial for determining egg quality, they do not seem central to the study's main purpose.

As suggested, we removed the 4th point.

  1. L111: Does the symbol "S" refer to "supermarket" or indicate the size of the purchased eggs? If the former, what was the size of the commercial eggs purchased, and how does it compare to the size of eggs laid by backyard hens?

The symbol “S” refers to “supermarket” while the symbol “R” refers to “rural”

Other comments:

L49-50: Consider removing this sentence. A non-technical simple summary would be more appropriate.

We removed this sentence.

L103: Does "±12 months" intend to mean "at the age of approximately 52 weeks"?

Checked and modified

Be consistent and use the term "eggshell" throughout the study.

Done

L122: Correct the Haugh unit formula to "w^0.37".

Done

L134: Correct the reference format.

Done

For all equipment used, provide the apparatus model number. For equipment and chemicals, include the manufacturer's location. This is mandatory for this journal as it aids in replicating measurements.

Done

L153: This line is unclear; please revise.

Done

L154: There is no isotope as "hg2O2". Carefully revise the isotope notation – Pb208 vs 75As.

Done

L162: Change "part number" to "code"; standards are not part of the apparatus. Also, this standard appears to have a concentration of 10 μg/mL, not 1 μg/mL; please verify.

The concentration used is not an error; it corresponds to the standard diluted with the solution mentioned above. We have improved the clarity of the text.

The title of Table 6 is misleading, as no comparison with other studies is provided.

As suggested, we modified the title of Table 6.

L303-325: Try to shorten this paragraph. Suggestion: Summarize the toxicological effects and exposure limits of all analyzed toxic elements in a table.

We have shortened the paragraph and moved it to the 'Introduction' chapter.

Round 2

Reviewer 1 Report

Comments and Suggestions for Authors

Dear authors, please find my comments in attached file. There are few more things that need to be resolved.

Best regards

Author Response

Dear authors, please find my comments in attached file. There are few more things that need to be resolved.

Best regards

Reply: As requested, the changes have been made following your comments.

Tables 2 and 3 were checked and corrected.  Superscript letters have been checked and ordered with the same criteria in all tables.

The data in Table 6 have been corrected as suggested. I apologise for the error.

In the reference list, n. 5 corresponds to Waegeneers et al., 2009. Is think it is correct.

Reviewer 2 Report

Comments and Suggestions for Authors

The title and objectives in the abstract make people think that this research is an overall study to represent ethically raised eggs. However, the whole results were based on one farm, which could be very likely not representative or even biased. 

Author Response

The title and objectives in the abstract make people think that this research is an overall study to represent ethically raised eggs. However, the whole results were based on one farm, which could be very likely not representative or even biased. 

Reply: Dear reviewer, as requested by you and another reviewer, the title, abstract and conclusion section were revised and implemented as requested.

Reviewer 3 Report

Comments and Suggestions for Authors

Thank the authors for making revisions. I would still like those issues I mentioned, which were not able to be changed, to be included in a condensed form as limitations of the study, as they limit comparisons between eggs layed by hens of different genotypes. This concerns the following aspects: the absence of analysis of toxic elements in the feed of backyard hens and the non-uniform age of the hens.

Author Response

Thank the authors for making revisions. I would still like those issues I mentioned, which were not able to be changed, to be included in a condensed form as limitations of the study, as they limit comparisons between eggs layed by hens of different genotypes. This concerns the following aspects: the absence of analysis of toxic elements in the feed of backyard hens and the non-uniform age of the hens.

Reply:  Dear reviewer, as suggested we revised the conclusions section, improving with the potential limitations of the study. We added lines 487-496.